# Longitudinal quantification of *Bifidobacterium longum* subsp. *infantis* reveals late colonization in the infant gut independent of maternal milk HMO composition

Dena Ennis [1], Shimrit Shmorak[1], Evelyn Jantscher-Krenn [2] & Moran Yassour [1,3] ✉

Breast milk contains human milk oligosaccharides (HMOs) that cannot be digested by infants, yet nourish their developing gut microbiome. While *Bifidobacterium* are the best-known utilizers of individual HMOs, a longitudinal study examining the evolving microbial community at high-resolution coupled with mothers' milk HMO composition is lacking. Here, we developed a high-throughput method to quantify *Bifidobacterium longum* subsp. *infantis (BL. infantis)*, a proficient HMO-utilizer, and applied it to a longitudinal cohort consisting of 21 mother-infant dyads. We observed substantial changes in the infant gut microbiome over the course of several months, while the HMO composition in mothers' milk remained relatively stable. Although *Bifidobacterium* species significantly influenced sample variation, no specific HMOs correlated with *Bifidobacterium* species abundance. Surprisingly, we found that *BL. infantis* colonization began late in the breastfeeding period both in our cohort and in other geographic locations, highlighting the importance of focusing on *BL. infantis* dynamics in the infant gut.

Breast milk is considered the ideal nutrition for infants during their first 6 months of life[1]. Millions of years of evolution have shaped breast milk composition such that its third most abundant component, human milk oligosaccharides (HMOs), cannot be digested by the infant, but serves as substrate for the infant's gut bacteria[2,3]. There are many different types of HMOs, which can be largely classified into three groups: fucosylated, sialylated, or neutral. Each HMO is composed of 3 to 32 monomers, and a single milk sample typically contains 50 to 200 distinct types of HMOs[4]. Among other factors, maternal genetics plays a role in the production of specific HMOs in breast milk[5].

For example, mothers with an inactive fucosyltransferase 2 (*FUT2*) gene, termed non-secretors, fail to form alpha-1,2 bonds between fucose and lactose or other HMO backbone structures, resulting in the lack of 2'FL and other alpha-1,2-fucosylated glycans[6]. Additionally, environmental factors coupled with infant age can affect HMO composition[7].

On average, infants have a higher relative abundance of *Bifidobacterium* species while they are breastfed[8–10]. *Bifidobacterium* species were previously shown capable of utilizing multiple HMOs[11–13], however this ability varies between species and even within a single

[1]Microbiology & Molecular Genetics Department, Faculty of Medicine, The Hebrew University of Jerusalem, Jerusalem, Israel. [2]Department of Obstetrics and Gynecology, Medical University of Graz, Graz, Austria. [3]The Rachel and Selim Benin School of Computer Science and Engineering, The Hebrew University of Jerusalem, Jerusalem, Israel. ✉e-mail: moranya@mail.huji.ac.il

species[14–17]. Among all *Bifidobacterium* species and subspecies, the best-known HMO utilizer is *Bifidobacterium longum* subsp. *infantis* (*BL. infantis*) which grows efficiently on most types of HMOs[18], and possesses a large variability of HMO utilizing genes[19]. In contrast, other *Bifidobacterium* species have a lower capability of HMO utilization, for example *B. breve* strains cannot utilize 3'SL and 6'SL at all, and most of them cannot utilize fucosylated HMOs[15]. Since HMOs serve as food for the gut microbiome, one may hypothesize that different HMO compositions in mothers' milk affects the developing gut microbial community.

To date, most research addressing the HMO-bacteria relationship in the infant gut focused on a single time point[20]. A longitudinal cohort study is needed in order to examine how changes in HMO composition impact the infant gut microbiome over time.

To quantify the abundance of various *Bifidobacterium* species in microbiome communities two approaches are commonly used: 16S-rRNA sequencing and shotgun metagenomics. While metagenomic sequencing allows classification at the species level, 16S-rRNA sequencing provides only genus-level classification of microbiome communities. The basic annotated unit in 16S-rRNA sequencing is referred to as operational taxonomic unit (OTU), which can be assigned to genus-level classification and may represent multiple species of the assigned genus. Multiple species can be annotated as the same OTU, hence using 16S-rRNA sequencing so far provided mostly weak or no associations with abundances of specific HMOs[21–24]. A single OTU can include multiple species (or subspecies) with various HMO utilization capabilities, thus, a higher-resolution taxonomic definition is needed.

The largest variability in HMO-utilization capability can be found within the *Bifidobacterium longum* species. Overall, this species can be divided into two subspecies found in humans: *B. longum* subsp. *longum* (*BL. longum*) and *B. longum* subsp. *infantis* (*BL. infantis*). *BL. longum* is found both in infants and adults, while *BL. infantis* is unique to the infant gut. Studies have shown that *BL. infantis* can utilize almost all HMOs[25], while *BL. longum* has a limited repertoire. To study the HMO-microbe relationship, taking into account these differences in HMO-utilization within *B. longum* subspecies, a high-throughput, higher-resolution method is needed. Past studies have used different methods to differentiate between *BL. infantis* and *BL. longum*, such as qPCR[26], PCR[27] or the Bifidobacterium Longum-Infantis Ratio (BLIR) method[28,29], yet these methods require the original DNA and are not high-throughput. Others have searched for *BL. infantis* specific genes such as the H1 cluster[20] or other *BL. infantis* clusters[30,31], however these methods do not give an exact ratio between the subspecies, rather they indicate their presence or absence. The new method we propose here could be applied also to the massive amounts of data available in public repositories.

Here, we establish a new matched cohort of breast milk and infant stool samples collected longitudinally throughout the first year of life. We develop a method to allow *BL. infantis* quantification from existing metagenomic data, and apply it to samples from our cohort to study the relationship between the abundance of *Bifidobacterium* species in the infant gut and HMO composition in mothers' milk over time. Finally, we apply our *B. longum* subspecies quantification method to existing infant gut datasets to examine the timing of *BL. infantis* colonization across geographic locations.

## Results
### Cohort design
We have established a new and unique longitudinal cohort to test the relationship between HMOs in mothers' milk and the developing infant gut microbiome. Our cohort consists of 21 mother-infant dyads with matched infant stool samples and breast milk samples collected on the same day. Altogether, we collected 80 stool samples and 50 breast milk samples together with the infant nutritional

information and antibiotic treatments (Supplementary Fig. 1, Supplementary Data 1). We collected these samples between the age of 2 weeks and 41 weeks, and each dyad contributed between one to eight paired samples.

### Specific marker genes allow better quantification of *B. longum* subspecies
*Bifidobacterium longum* subsp. *infantis* (*BL. infantis*) is the best known utilizer of HMOs[19,32], however current methods for taxonomic classification from metagenomes are unable to separate the *Bifidobacterium longum* (*B. longum*) species into its main subspecies; *Bifidobacterium longum* subsp. *longum* (*BL. longum*) and *BL. infantis*[20,29]. MetaPhlAn is one of the most common tools for profiling the composition of microbial population from metagenomic data, by using specific marker genes for each taxonomic group[33]. However MetaPhlAn has no specific marker genes for *BL. infantis* and therefore classifies *B. longum* at the species taxonomic level. Due to the differences between *B. longum* subspecies in the context of HMO utilization, there is a rising need for a high-throughput method that will allow specific identification and quantification of *BL. infantis* from metagenomics data.

Here we define *B. longum* subspecies specific markers and use them in a tailored MetaPhlAn[33] database which allows abundance quantification of two *B. longum* subspecies: *BL. infantis* and *BL. longum* (Supplementary Fig. 2A). To construct our new dataset, we searched for marker genes that are unique to each subspecies. A marker gene was selected if it was present in at least 90% of reference genomes of one subspecies and not in a single genome of the other subspecies (Fig. 1A, Methods). We chose to discard two subspecies of *B. longum*: *Bifidobacterium longum* subsp. *suis* and *Bifidobacterium longum* subsp. *suillum* since they are rarely found in humans[34], and a limited amount of reference genomes exist for these subspecies (Methods).

In order to validate our results, we applied MetaPhlAn with our new set of marker genes coupled with subspecies-specific qPCR to metagenomic sequencing data from 68 infant stool samples. When comparing the relative abundance of *BL. infantis* and *BL. longum* in each method, we observed a strong correlation between our computational approach and qPCR ($R^2 = 0.999$ for *BL. infantis* and 0.997 for *BL. longum*; Fig. 1B, C). This finding confirms our method's specificity and sensitivity for both *BL. infantis* (Fig. 1B) and BL. *longum* (Fig. 1C). In some samples, MetaPhlAn failed to assign a classification to a small percentage of *BL. longum* and therefore it was designated as unclassified *B. longum* (Supplementary Fig. 2B, C).

### The infant gut microbiome shows excessive changes while HMO composition in mothers' milk is fairly stable
To examine the dynamics of *Bifidobacterium* species in the infant gut, we conducted metagenomic sequencing and analyzed the data using our novel MetaPhlAn database. We observed a significant prevalence of *Bifidobacterium* (at least one sample with >25%) in all the infants (Fig. 2A), in line with our expectation as most infants in our study were breastfed[8,14,35]. *Bifidobacterium* remained highly abundant even after solid foods were introduced to infants (Fig. 2A, arrows). The abundant species included *Bifidobacterium breve*, *Bifidobacterium bifidum*, *Bifidobacterium pseudocatenulatum* and *B. longum* subspecies (Fig. 2A), along with *Bacteroides* species such as *Bacteroides dorei* and *Bacteroides vulgatus* (Supplementary Fig. 3). Interestingly, the presence of *BL. longum* and *BL. infantis* was mutually exclusive, reflecting potential intra-species competition, as previously suggested[36].

Overall, we found that the infant gut microbiome underwent significant changes over the course of several months. While the bacterial composition tended to be stable over a few weeks, there were certain time points when a switch in composition occurred (Fig. 2A, B). In infants that we had samples from many time points we found this switch to occur around 10 weeks of life. For example, in

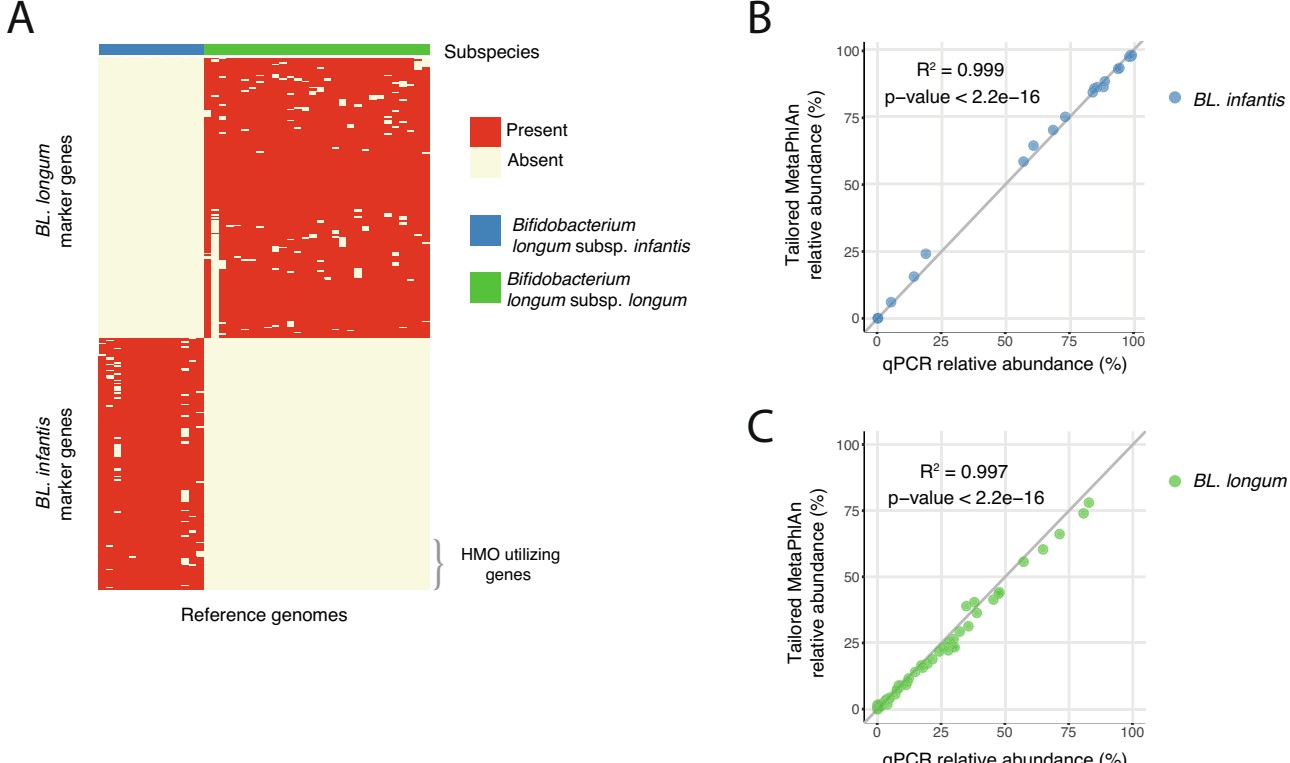

**Fig. 1 | Quantification of *Bifidobacterium longum* subspecies using marker-genes enables subspecies-level detection. A** Identification of unique genes in *BL. infantis* (blue) and *BL. longum* (green) reference genomes that can serve as marker-genes in the quantification of *B. longum* subspecies. HMO utilizing genes (HUGs) are marked on the bottom right and constitute only a small fraction of *BL. infantis* marker genes. **B**, **C** Validation of the computational approach, comparing the tailored MetaPhlAn marker-gene quantification results to the experimental qPCR results of **B** *BL. infantis* and **C** *BL. longum*. *P*-value for regression was calculated using a two-sided *t*-test.

infant03 (inf03) over the course of the first 15 weeks the dominant bacteria in the gut constantly changed, including *B. vulgatus*, *Klebsiella pneumoniae*, and finally *B. breve*; the gut of infant18 was initially dominated by *B. breve*, followed by a complete switch to *BL. infantis*; and the gut of infant16 constantly changed its microbial composition (dominated by *Escherichia coli*, *Veillonella seminalis* and *B. pseudocatenulatum*; Fig. 2B). It was not always clear what triggered these microbial shifts, however since *Bifidobacterium* and other gut microbes utilize HMOs, we hypothesized that changes in the HMO composition in mothers' milk might be causing bacterial changes in the infant gut.

To examine the impact of HMO composition in mothers' milk on the infant gut microbiome, we quantified 16 common HMOs in 50 milk samples from 20 mothers using high performance liquid chromatography with fluorescence detection (HPLC-FLD; Methods). In contrast to the dynamic infant gut microbiome, the composition of HMOs in mothers' milk remained relatively stable over the course of months, in terms of both their concentration in milk (Supplementary Fig. 4A) and the relative abundance of specific HMOs (Fig. 2C). We divided the milk samples into three main groups, based on their HMO composition: those with low or no 2'FL abundance (Group 1, non-secretors[37]); samples from secretor mothers with very high abundance of 2'FL (>40%) and LNFP1 (>10%) and smaller amounts of other HMOs (Group 2); and samples from secretor mothers with lower abundance of 2'FL (<30%), and no clear dominant HMO (Group 3; Fig. 2C). We found no major changes over time in the abundance of specific HMOs, other than LSTc which was reduced to almost 0 over the course of ~40 weeks (Supplementary Fig. 4B, C), in line with previous findings[7]. Overall, the changes found in the HMO composition in consecutive milk samples were significantly less pronounced than those found in the microbial

population from consecutive infant gut samples (*t*-test, *p* = 2.88e-5, Supplementary Fig. 5A).

## The dominant *Bifidobacterium* species shape the infant gut microbiome

General comparison of the infant gut microbiome composition determined that *Bifidobacterium* species play a significant role in the breastfed infant gut (Fig. 2A). We next searched for differences across samples in an attempt to characterize the various microbial profiles of the infant gut. We examined the diversity of the infant gut samples in our cohort, using a dimension reduction approach (PCoA with Bray-Curtis dissimilarity, Methods), and found that samples cluster into distinct clusters (using K-means, *k* = 4). The first three groups had samples with mostly one main *Bifidobacterium* dominant species (with relative abundance of >30%): *B. breve*, *BL. longum* and *BL. infantis*, and the fourth group contained samples dominated by either a different *Bifidobacterium* species (*Bifidobacterium adolescentis*, *B. pseudocatenulatum*) or other species (named "Mixed", Fig. 3A). While usually consecutive samples from the same infant were assigned to the same cluster, occasional cluster switches were observed, strengthening our finding that the microbiome changes in this time frame (Fig. 3B). Overall, these analyses highlight the importance of *Bifidobacterium* species in our samples, as these are major factors that impact the variation in the infant gut microbial composition.

To further investigate these four groups of infant gut samples, we examined the alpha diversity of the microbiome population in samples found in each group using Shannon index. We found that the alpha diversity of samples within the *BL. infantis* group was lower compared to samples from other groups (*t*-test, *p* ≤ 0.001, Fig. 3C), indicating that when *BL. infantis* is found, it dominates the

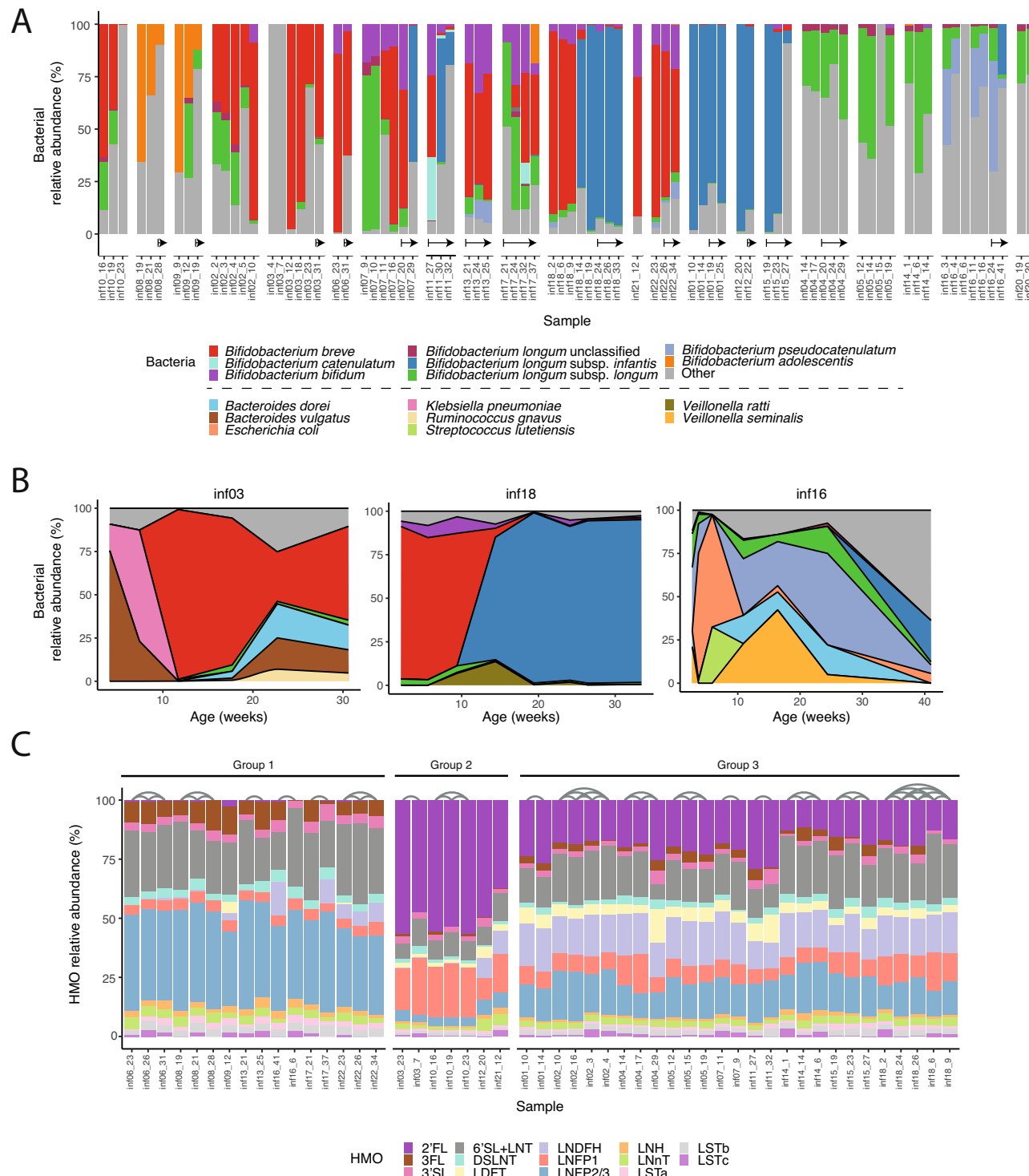

**Fig. 2 | Composition and dynamics of *Bifidobacterium* species in the infant gut and HMOs in mothers' milk. A** Relative abundance of *Bifidobacterium* species in the infant gut (colorful) with all other species classified as "other" (gray). Samples from the same infant are grouped together, sorted by age. Arrows indicate samples taken after the introduction of solid food. **B** Temporal changes in the relative abundance of the microbial community in three infants (inf03, inf16, and inf18) in

the first 30–40 weeks of life. The most prevalent bacteria are colored, and the remaining are indicated as "Other" (gray). Colors as in (**A**), with additional colors for the non-*Bifidobacterium* species. **C** Relative abundance of 16 HMOs measured in mothers' milk. Samples are categorized into three groups based on their HMO profiles, and arches connect samples obtained from the same mother (Methods).

community at such high levels of relative abundance leaving a smaller ecological niche for other bacteria in the infant gut. We next focused on the "Mixed" group, and asked whether additional variables may play a role in these microbial profiles. We examined breastfeeding, maternal or infant antibiotic use, delivery mode,

breastfeeding type (pumped or direct), and introduction of solid foods, yet we did not find any specific variable that was associated with the microbial profile of the "Mixed" group. As expected, the *BL. infantis* group consisted solely of infants who received none-to-low amounts of infant formula[38].

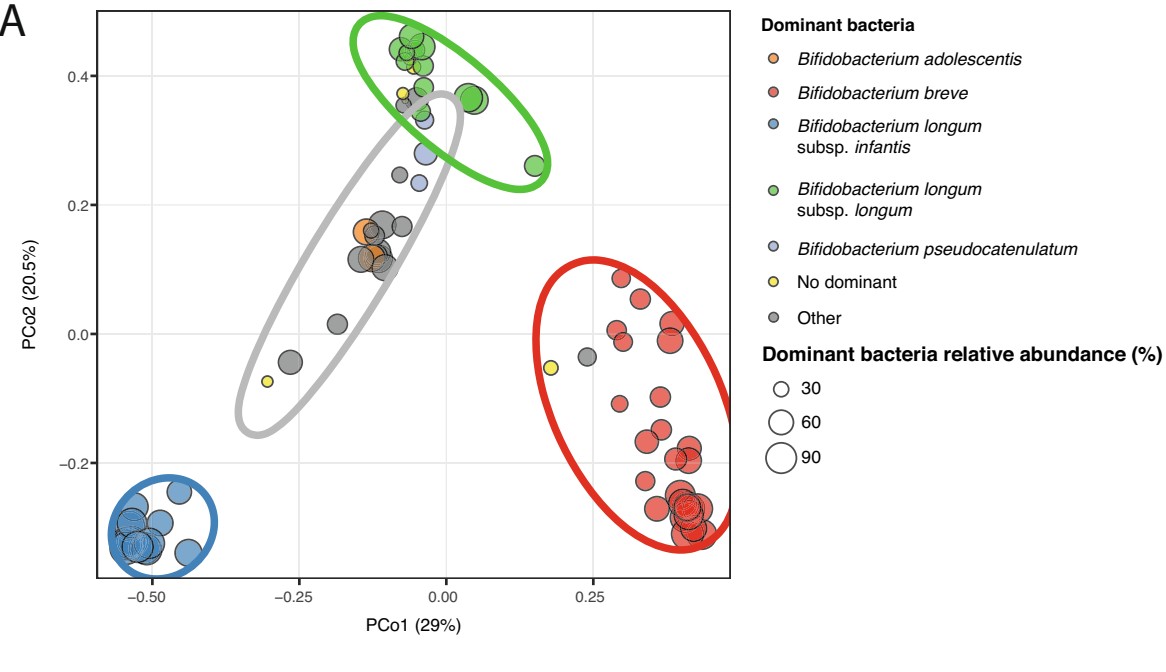

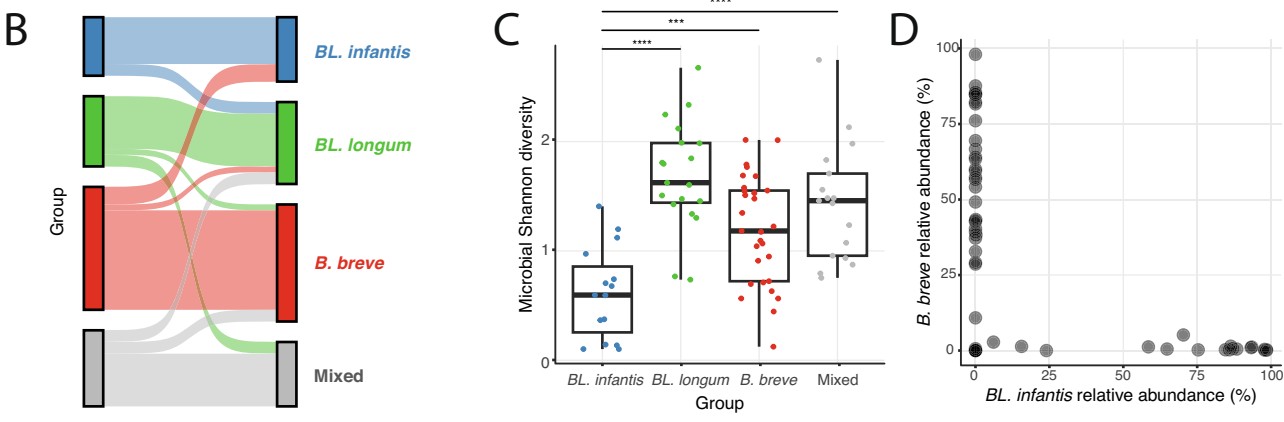

**Fig. 3 | Variation across samples is highly impacted by the dominant *Bifidobacterium* species. A** Principal Coordinate Analysis (PCoA) of infant gut microbiome samples using Bray-Curtis dissimilarity. Points are color-coded based on the dominant *Bifidobacterium* species in each sample, and non-Bifidobacterium dominated samples are colored in gray. Samples without a dominant species (>30% relative abundance) are labeled as "No dominant" (yellow). The size of each point represents the relative abundance of the dominant bacteria in that sample. Ellipses encompass four groups identified using k-means clustering. Each group represents a primary dominant species (indicated by the ellipse color): *BL. infantis* (blue), *BL. longum* (green), *B. breve*

(red), and "Mixed" (gray). **B** Changes in group assignment observed in consecutive samples from the same infant (colors as in A). **C** Microbial diversity of samples within each group (measured by the Shannon index; colors as in A; two sided *t*-test, $n = 80$ samples from 21 infants, ***$p \le 0.001$, ****$p \le 0.0001$). Box boundaries are the 25th and 75th percentiles, and the median is highlighted. Whiskers represent 1.5 * IQR and points past them are outliers. **D** Relative abundance of *BL. infantis* (x-axis) versus the relative abundance of *B. breve* (y-axis) in each sample, indicating the mutual exclusiveness of the two species.

To characterize the relationships among dominant *Bifidobacterium* species in the infant gut, we examined their occurrence within groups where they are not dominant. We found that *BL. infantis* and *B. breve* are mutually exclusive, consistent with a previous study in Hazda infants[39], implying competition for the same niche in the infant gut (Fig. 3D). However, it remains unclear what specific niche *B. breve* and *BL. infantis* are competing for, given *B. breve*'s limited ability to utilize a variety of HMOs[15]. Finally, we observed *BL. longum* in some *B. breve*-dominant samples (Fig. 2A), suggesting that *B. breve* may rely on derivatives from *BL. longum* through cross-feeding in these samples[40].

## Single HMOs are not associated with specific *Bifidobacterium* species

It is well established that different *Bifidobacterium* species have different HMO-utilization capabilities[14,40,41]. Therefore, specific HMOs may benefit specific *Bifidobacterium* species in the infant gut based on their HMO utilization profiles. However, we found no significant correlation between the abundance of *Bifidobacterium* in general and the main *Bifidobacterium* species and subspecies with specific HMOs (Fig. 4A) or HMO groups (fucosylated, sialylated or neutral; Supplementary Fig. 5B). In addition, linear association models accounting for individual infants showed no significant association between specific

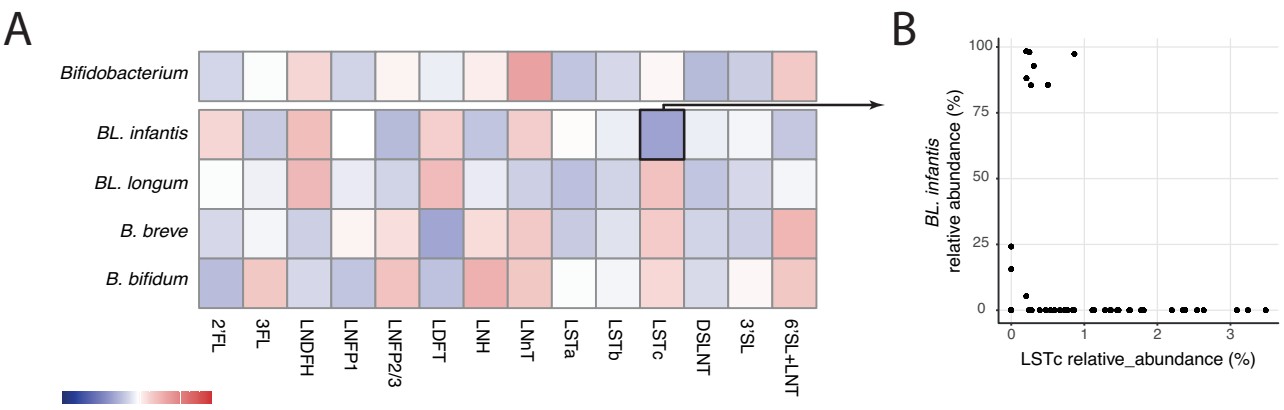

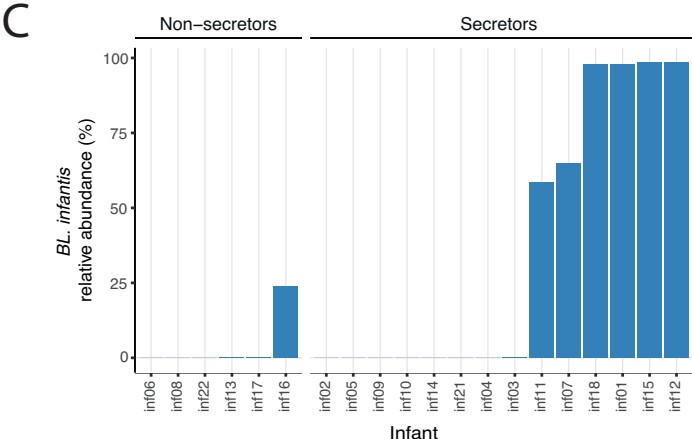

**Fig. 4 | Mothers' milk HMO composition shows no signieficant correlation with infant gut *Bifidobacterium*. A** Pearson correlations between all measured HMOs and the *Bifidobacterium* genus, as well as the main individual species and sub-species (*BL. infantis*, *BL. longum*, *B. breve* and *B. bifidum*). None of these correlations were found to be statistically significant (*t*-test). **B** Comparison of the relative abundance of LSTc in mothers' milk (*x*-axis) with the relative abundance of *BL. infantis* in the infant gut microbiome (*y*-axis). **C** Maximum relative abundance of *BL. infantis* in each infant divided into two groups of infants with secretor and non-secretor mothers, highlighting the high abundance of *BL. infantis* only in infants to secretor mothers, but not in all of them.

HMOs and *Bifidobacterium* species and subspecies in the infant gut (when requiring FDR *q* < 0.2). Nevertheless, we found that *BL. infantis* exhibited a high abundance (>25%) exclusively in infants to secretor mothers (Fig. 4B, *t*-test *p* = 0.021). In addition, we observed a modest and non-significant negative correlation (*r* = −0.27) between *BL. infantis* and LSTc (Fig. 4C). It is worth noting the importance of considering the timing factor in interpreting these findings. The delayed presence of *BL. infantis* in the gut (which will be discussed in more detail later) and the gradual decrease of LSTc over time (Supplementary Fig. 4B,C) could contribute to the observed correlation. The lack of variation in the HMO composition together with the lack of HMO-microbes associations indicate that the microbial shifts, specifically within *Bifidobacterium* species, can not be explained by a change in mothers' milk HMO composition.

## Metagenomes with *BL. infantis* contain more HMO utilizing genes

Metagenomes obtained from various time points of multiple infants contain distinct strains and species, resulting in variable gene abundance profiles which can enable various patterns of HMO utilization. To assess the HMO utilization potential of specific *Bifidobacterium* species in our dataset, we investigated the presence of HMO-utilizing genes (HUGs)[42] organized into five distinct clusters (H1-H5[32]; Fig. 5). We observed that the dominant species in each sample significantly

influenced the metagenome's theoretical capacity for HMO utilization. As expected, *BL. infantis*-dominated samples exhibited the highest abundance of HUGs, confirming its exceptional capability in utilizing HMOs[19]. Notably, some of these samples displayed high variation in gene abundance from clusters H1 and H5, indicating a potential lower capacity to transport some HMOs[19], and utilize lacto-N-tetraose (LNT) and lacto-N-neotetraose (LNnT)[16].

Interestingly, metagenomes that were dominated by *B. breve* or *BL. longum* also contained genes from the H5 cluster, emphasizing their ability to utilize HMOs based on lacto-N-biose (LNB)[16]. However, samples from the "Mixed" groups exhibited minimal or no HUGs, suggesting either alternative genes for HMO utilization or a lack of capacity to utilize HMOs altogether.

## *BL. infantis* does not colonize the infant gut in early breastfeeding weeks

*BL. infantis* is the most proficient HMO-utilizer in the infant gut[11,19], thus we expected that *BL. infantis* will have a fitness advantage in the breastfed infant gut from the initial days of breastfeeding. However, despite the majority of infants in our cohorts that were breastfed since birth, *BL. infantis* was primarily detected starting only at 10 weeks of age (Fig. 6A). Linear association models showed a clear positive association of the relative abundance of *BL. infantis* with age (coefficient = 3.16, *q* = 6.63e-5), which was not found for any other

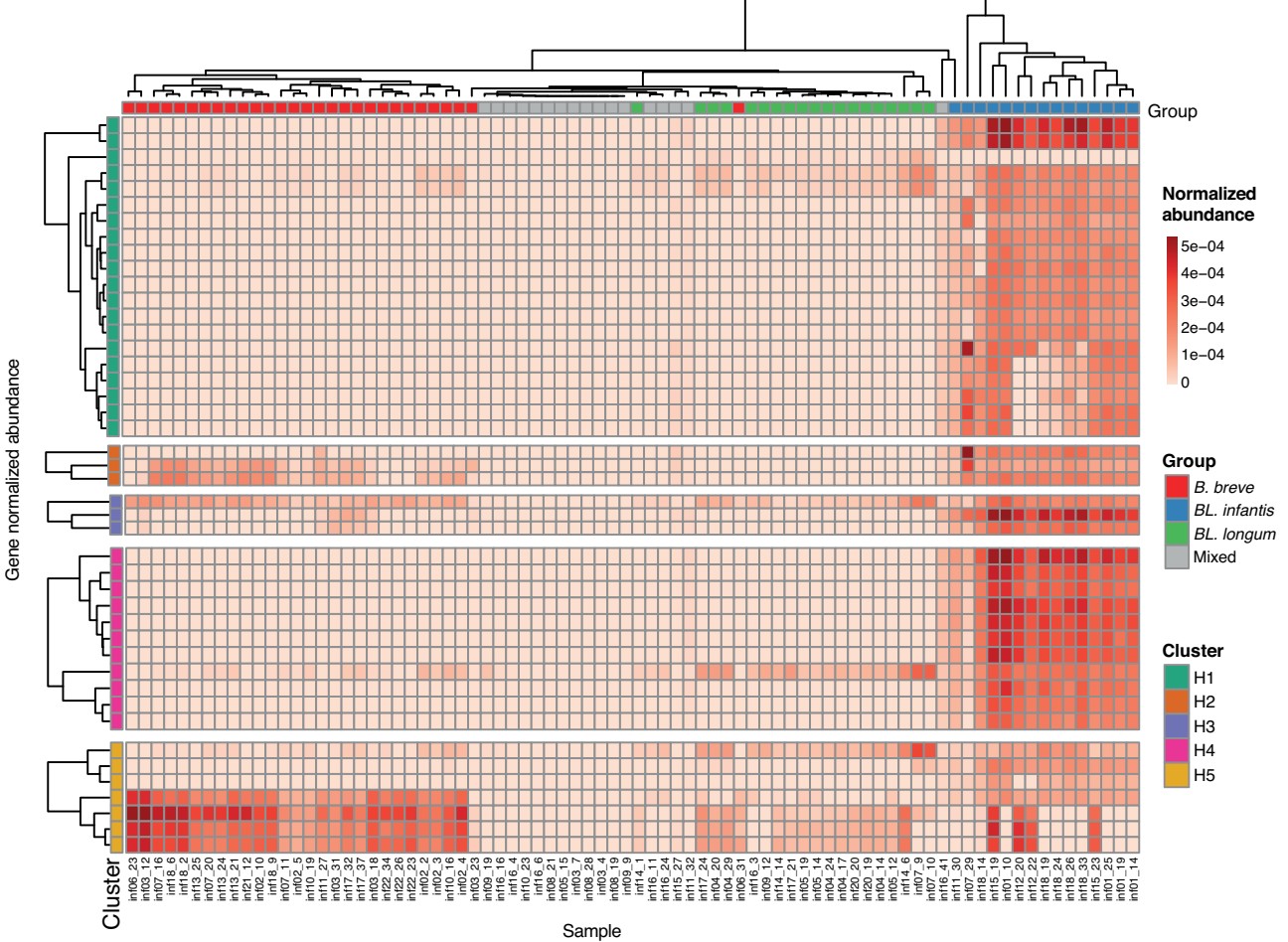

**Fig. 5 | Abundance of HMO utilization genes in metagenomic samples.** Normalized genes abundance of HMO utilization genes (HUGs) in all infant samples. Genes (rows) are categorized and labeled according to their respective gene cluster (H1-H5). Samples (columns) are annotated based on their assigned dominant-species group.

*Bifidobacterium* species or subspecies. Overall, *BL. infantis* exhibited the highest abundance at 10–25 weeks, followed by a gradual decrease in abundance (Fig. 6A).

Our innovative computational methodology enabled the exploration of the delayed colonization of *BL. infantis* across various geographical regions, providing a gateway for further in-depth investigation into this phenomenon. Thus, to corroborate our findings, we examined additional infant cohorts from Sweden[9], United Kingdom[43], Estonia[44], Italy[45], Russia[44], United States[46] and two cohorts from Finland[44,47] comprising samples from a total of 1,017 infants throughout the first year of life. Across all cohorts, a similar pattern of late-colonization of *BL. infantis* was observed: In Sweden, *BL. infantis* was not observed at birth almost at all, reaching its peak prevalence at 17 weeks, followed by a gradual decline in both prevalence and relative abundance by 52 weeks (Fig. 6B). In the UK, *BL. infantis* was observed in only four out of 178 infants in the first 3 weeks of life, and during the later infancy period (17–52 weeks) *BL. infantis* was found in 28 infants (Fig. 6C). In Finland, one cohort with 126 infants had only a single infant with detectable levels of *BL. infantis* in the first 10 weeks of life, and additional 11 infants gained it later on (Fig. 6D). In a second Finnish cohort, *BL. infantis* was not observed in any infant samples (ages ≤ 13 weeks; Finalnd2, Supplementary Fig. 6A). In the Italian and Estonian cohorts, a similar pattern was observed however the relative abundance of *BL. infantis* was lower in most infants (Supplementary Fig. 6B, C). The cohort from the the United states contained samples only from the first 2 weeks of life, revealing no infants with *BL. infantis* in the first

week of life, and only one out of 77 infants with detectable levels in the second week of life (Supplementary Fig. 6D). Finally, in the Russian cohort which consisted of samples collected from infants aged 12 weeks and beyond, *BL. infantis* was detectable in 14/69 infants,with its presence becoming noticeable only at 20 weeks (Supplementary Fig. 6E). Overall, all cohorts exhibited a late-colonization of *BL. infantis*, commonly starting at 10 weeks of age, or later (paired *t*-test, *p* = 0.023; Fig. 6E, Supplementary Fig. 6F).

To examine variations between *BL. infantis* strains across the different countries we next focused on *BL. infantis* strain-level composition, using the SNP profiles on the newly-identified *BL. infantis* marker genes (Methods; Fig. 6F). We found that *BL. infantis* strains within the same infant in our cohort were more similar to each other than strains between unrelated infants (*t*-test, *p* ≤ 2.2e-16, Supplementary Fig. 6G). Furthermore, in some countries (i.e., Italy & Russia), most of the strains were very similar, while in other countries (i.e., Sweden & UK) there was a much larger variation between strains (Fig. 6F, Supplementary Fig. 6H). Strains from our Israeli cohorts were clustered in two distinct groups, one more similar to strains found in the US (light purple, Fig. 6F) and the other closer to strains found in the UK (brown, Fig. 6F). Specifically, the strains from the Italian samples were very distinct from all other strains, with the exception of a single UK strain. Finally, *BL. infantis* reference genomes were clustered into three groups, some identical to the Italian strains, while others clustered closely with Russian and Estonian strains (gray, Fig. 6F). These findings allow us to explore the variation found within *BL.*

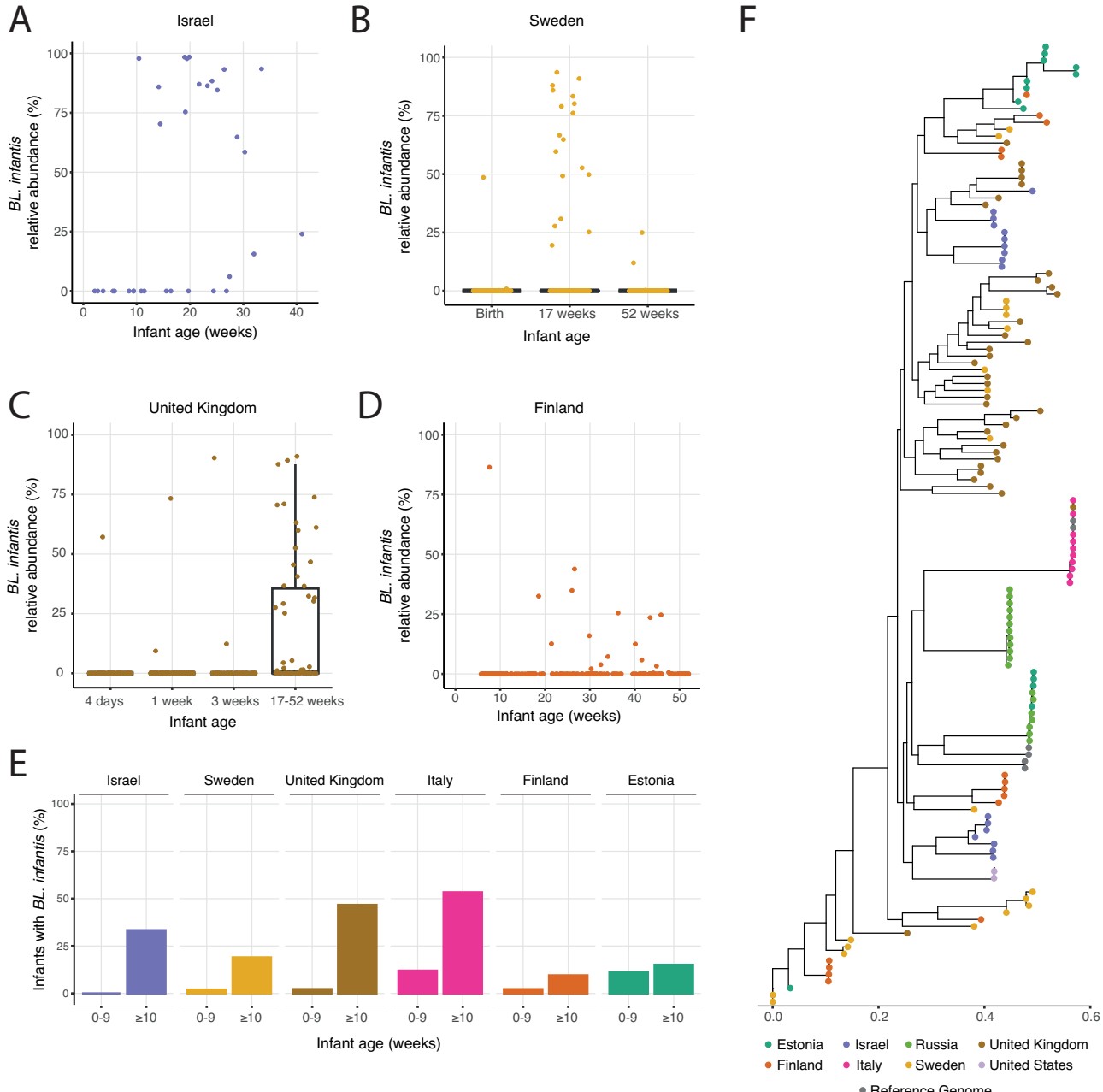

**Fig. 6 | Late colonization and low prevalence of *BL. infantis* in the infant gut in multiple geographic locations. A** Relative abundance of *BL. infantis* in our Israeli cohort, among infants where *BL. infantis* was detected at any time point. **B**–**D** Relative abundance of *BL. infantis* at different time points in samples from **B** Sweden (*n* = 300 samples from 100 infants) **C** UK (*n* = 376 samples from 169 infants) and **D** Finland (*n* = 228 samples from 108 infants). For Sweden and the United Kingdom where samples were collected at defined time points box plots are shown, with box boundaries at the 25th and 75th percentiles, and the median highlighted. Whiskers represent 1.5 * IQR and points past them are outliers. **E** The percentage of infants in multiple geographical locations harboring *BL. infantis* in the first 10 weeks of life and at 10 weeks and later. **F** Phylogenetic tree displaying *BL. infantis* strains found across all cohorts. The dominant strain from each sample is represented, and strains are color-coded based on their corresponding cohort, highlighting that some countries (like Italy) have very similar strains in all samples.

*infantis* and highlight interesting variability across geographic locations.

## Discussion

In this study we introduced an innovative approach to quantify *BL. infantis* and distinguish it from *BL. longum* in metagenomic data. Our method enables researchers to concentrate on studying this distinct subspecies and its associations with HMOs from existing metagenomic data. This approach can be adapted to differentiation of subspecies in additional microbial species, and specifically further within *B. longum* (such as *BL. suis* & *BL. suillum* subspecies) once sufficient numbers of

reference genomes become available. We employed our approach to explore the diversity within the infant gut microbiome and discovered the lack of associations with individual HMOs present in mothers' milk. Our analysis revealed that the variability between samples was greatly influenced by the dominant *Bifidobacterium* species in each sample.

Previous research has suggested that colonization of *Bifidobacterium* species in the infant gut may be influenced by priority effects[40]. However, our study revealed substantial changes in the dominant *Bifidobacterium* species within the same infant over the course of several weeks (Fig. 3A, B). This indicates that over time there are additional factors responsible for *Bifidobacterium*

species prosperity, such as species competition and cross-feeding. For example, it was reported that *B. breve*, despite having limited ability to utilize HMOs, can outcompete stronger competitors if introduced early into a microbial community[40]. In addition, *B. breve* has the capacity to cross-feed on monosaccharides derived from HMOs by other *Bifidobacterium* species[48,49]. This implies that *B. breve* may initially dominate the population when carbohydrates are available, but subsequently loses the competition to other *Bifidobacterium* species once these carbohydrates are depleted. In our data we found that the *B. breve* dominated group had a diverse microbial population (Fig. 3C), perhaps since it cross feeds on HMOs derivatives from other species. More research is needed to understand the microbial shifts in the infant gut and the effect cross-feeding has on the microbial dynamics in the infant gut.

While the HMOs present in a mother's milk remained relatively stable over numerous weeks, we observed notable changes in the infant gut microbiome during this time period. This suggests that even subtle variations in the composition of breast milk may have an impact on the development of the gut microbiome. Alternatively, it is possible that other components present in breast milk, such as cytokines, microRNAs and antibodies, play a role in influencing the infant's microbiome[50,51]. In addition, we can not rule out environmental effects that may contribute to these changes such as starting daycare which in Israel typically occurs at 15–26 weeks of age. Our findings indicate that there were no significant correlations between HMOs and *Bifidobacterium*, further supporting the idea that additional factors beyond HMOs are involved in shaping the infant's microbiome. Previous studies[22–24,26] have examined the composition of the microbiome and its relationship with HMOs using 16S-rRNA amplicon sequencing. These studies have reported varying results, with some finding no significant correlations, while others observed modest correlations. Interestingly, some of the studies identified a negative correlation between *Bifidobacterium* OTUs and multiple HMOs[22,24]. This could be attributed to a decrease in the overall *Bifidobacterium* abundance within the infant gut over time, coupled with an increase in specific HMOs, in line with our findings regarding LSTc and *BL. infantis* (Fig. 4B). Additionally, it is possible that such specific correlations may be observable only using a larger cohort.

Finally, we found that *BL. infantis* does not commonly colonize the infant gut in the early weeks of breastfeeding and that not all breastfed infants have detectable levels of *BL. infantis*. Importantly, when *BL. infantis* was found in the infant gut, it commonly dominated the gut microbiome community. Analyzing additional infant cohorts from multiple geographical locations strengthened our findings regarding the late colonization of *BL. infantis* in infants' gut. Previous studies have reported a low prevalence of *BL. infantis* during early time points[29,36], however these studies did not incorporate frequent sampling in the first months of life, thus lacking the ability to precisely determine the timing of *BL. infantis* arrival. A previous study proposed that the arrival of *BL. infantis* is influenced by the history of breastfeeding practices of a given country[29], which impacts the overall exposure to *BL. infantis* strains. Countries with historically lower breastfeeding rates are likely to have a lower prevalence of *BL. infantis*, resulting in infants acquiring *BL. infantis* at a later stage through horizontal transfer[29].

Although *BL. infantis* is one of the more-studied infant gut commensals, its publicly-available reference genomes do not span the entire genomic variation of this subspecies (gray, Fig. 6F). To expand the *BL. infantis*-related research, additional reference genomes should be characterized in full, from natural isolates representing diverse geographic locations and lifestyles. Further research, especially longitudinal sampling of infants and their surroundings, is required to elucidate the timing and sources from which infants acquire *BL. infantis* and to comprehend the differences observed between countries.

## Methods

### Sample collection
Breast milk and stool samples were collected as part of the Breast Milk Baby (BMB) cohort from mothers and infants from birth till 1 year old. Stool samples were collected using eSwab® with 1 ml of liquid Amies medium + 1 regular FLOQSwabs® (Copan) in order to preserve bacterial population. Breast milk samples were collected by pump or manually and stored in sterile tubes. Both sample types were collected by mothers in their homes and stored at 4 °C for up to 24 h and then shipped to the lab and stored long term at −80°C.

### Metagenomic library construction and sequencing
DNA was extracted from stool samples using DNeasy PowerSoil Pro Kit (#47014, QIAGEN). Illumina sequencing libraries were prepared using Nextera XT DNA Library Preparation kit (FC-131-1096, Illumina) according to the manufacturer's recommended protocol with half of the volume and the DNA. Samples were sequenced using Illumina single-end 150 bp sequencing on a NextSeq 500 device.

### *B. longum* subspecies quantification
To identify *B. longum* subspecies-specific markers we started with 116 *B. longum* reference genomes downloaded from the NCBI with completeness of >90% and contamination of <5% (Supplementary Data 2). Reference genomes were classified to *B. longum* subsp. *longum* (*BL. longum*), *B. longum* subsp. *infantis* (*BL. infantis*) and unknown based on NCBI annotation, leaving 30 *BL. longum* and 16 *BL. infantis* references. As some of the unknown references could belong to either *BL. infantis* or *BL. longum*, we removed them all for specificity. We decided not to include subspecies *BL. suis* and *BL. suillum* in our method due to the limited availability of reference genomes for these subspecies leading to difficulty to generate reliable marker genes. In addition, these subspecies were rarely reported in humans[34], thus are less relevant in our settings. Therefore it is important to note that using our method, *BL. suis* and *BL. suillum* can be misidentified as a different subspecies or alternatively identified as *B. longum* species with unclassified subspecies.

PanPhlan3[52] was used to analyze the pangenome of all 46 reference genomes. Clustering all presence/absence profiles in the pangenome revealed two clear clusters, the first of 14 *BL. infantis* strains and the second included all *BL. longum* strains and two *BL. infantis* strains (*BL. infantis* 157 F, *BL. infantis* CCUG52486; Supplementary Fig. 7). Additional analysis showed that these two *BL. infantis* reference genomes that were clustered with *BL. longum* reference genomes did not contain the H1 HMO utilization cluster which defines *BL. infantis*[19]. In addition, a previous study showed that these two *BL. infantis* reference genomes are most probably *BL. longum*, based on the phylogeny of the core pangenome of 158 *B. longum* strains[53]. Taken together, *BL. infantis* 157 F and *BL. infantis* CCUG52486157 were suspected as mis-annotated and were excluded from further analysis.

Subspecies-specific marker genes were chosen in two steps. First, using the pangenome we found 331 genes that were present in 90% of one subspecies and not present at all in the other subspecies. For example, a gene that was present in 13 out of the 14 *BL. infantis* references and not in any *BL. longum* references was selected to be a marker gene for *BL. infantis*. Next, all selected genes were filtered to be specific at the species level to *B. longum* and to confirm they do not exist in other *Bifidobacterium* species. To do so, we used Blastn 2.12.0 to map all marker genes to the nr/nt nucleotide database. Of the 331 putative marker genes, 84 matched other species (such as *B. breve*) with >90% alignment and over 50% coverage and therefore were filtered out. Our final set of maker genes included 119 *BL. infantis* and 128 *BL. longum* markers (Supplementary Data 3). The MetaPhlAn database was customized to include the newly defined marker genes using described MetaPhlAn instructions (https://github.com/biobakery/MetaPhlAn/wiki/MetaPhlAn-4) and then MetaPhlAn4 was used with

the --index and --bowtie2db parameters and our customized marker-gene database.

To verify the results, subspecies of *B. longum* were determined using qPCR with subspecies-specific primers[54] for *BL. longum* (F: GTGTGGATTACCTGCCTACC, R: GTCGCCAACCTTGACCACTT) and *BL. infantis* (F: ATGATGCGCTGCCACTGTTA, R: CGGTGAGCGT-CAATGTATCT). The efficiency of the primers was assessed by testing them in five dilutions. qPCR was performed at 95 °C for 10 s, followed by 40 cycles of 95 °C for 10 s and 60 °C for 30 s. The ratio between *BL. infantis* and *BL. longum* was calculated using the delta-delta Ct method.

To use our tailored MetaPhlAn database see our GitHub page (https://github.com/yassourlab/MetaPhlAn-B.infantis/).

## Metagenomic analysis

Host reads were removed using an in house pipeline by aligning reads to the human genome by Bowtie2[55] (2.4.5-1). Samples were filtered and trimmed for Nextera adapters using fastq-mcf, ea-utils[56] (1.05). Taxonomic profiling was done using MetaPhlAn4[33] with our unique database as described above. Functional profiling was done using HUMAnN3[52]. HUGs[42] were selected from previously described HMO clusters[32]. Strain analysis was performed using StrainPhlAn 4[33] with default parameters and --sample_with_n_markers 50. SNPs for *BL. infantis* marker-genes were calculated using the multiple-sequence alignment (MSA) produced by StrainPhlAn 4 (--mutation rates parameter). *BL. infantis* was considered detectable when its relative abundance was ≥0.5%. Further analysis was done using an in house R (4.2.2) script utilizing dplyr[57] (1.1.2), tidyr[58] (1.3.0) and tidyverse[59] (2.0.0). Plots were created using ggplot2[60] (3.4.2) and ggforce[61] (0.4.1), colors were used from RColorBrewer[62] (1.1–3) and pals[63] (1.7). Heatmaps were created using pheatmap[64]. Alpha and beta diversity were calculated using "diversity" (Shannon index) and "vegdist" (Bray-Curtis dissimilarity) from the vegan[65] (2.6–4) package and the PCoA was created using the ape[66] (5.7-1) package. Phylogenetic tree was produced using ggtree[67] (3.6.2) and the sankey plots were created using ggsankey (0.0.99999). Additional cohorts were downloaded from NCBI Sequence Read Archive as following: Sweden[9] (PRJEB6456), United states[46] (PRJNA591079), Italy[45] (PRJNA352475), United Kingdom[43] (PRJEB32631), Finland, Estonia and Russia[44] (PRJNA497734) and an additional cohort from finland[47] (Finland2, PRJNA475246).

## HMO quantification

HMO standards used in this study were purchased from Dextra Laboratories, United Kingdom. These included 2'-fucosyllactose (2'FL), 3-fucosyllactose (3FL), 3'-sialyllactose (3'SL), 6'-sialyllactose (6'SL), lacto-N-tetraose (LNT), disialyllacto-N-tetraose (DSLNT), Lacto-difucotetraose (LDFT), lacto-N-difucohexaose 1 (LNDFH), lacto-N-fucopentaose (LNFP) 1, 2, and 3, lacto-N-hexaose (LNH), lacto-N-neotetraose (LNnT) and sialyl-lacto-N-tetraose (LST) a, b and c. Linear B6-Trisaccharide was used as an internal standard.

HMO quantification was performed as previously described[68,69]. Briefly 5 µl of human milk was combined with Linear B-6 Trisaccharide (Dextra Laboratories, UK) and HPLC grade water, then subjected to C18 columns (Thermo Scientific #60108-390) and carbograph columns (Thermo Scientific #60302-606) to remove proteins and salts respectively. Samples were labeled using 2-aminobenzamide (2-AB, Sigma) for 2 h at 65 °C. Excess 2-AB was removed using Silica columns (Thermo Scientific, #60300-482). Samples were separated by HPLC with fluorescence detection on a TSKgel Amide-80 column (Tosoh Bioscience, Tokyo, Japan) with a linear gradient of a 50 mM ammonium formate/acetonitrile solvent system. Retention times of purchased standard HMOs were used to annotate HPLC peaks. 6'SL and LNT peaks could not be separated and therefore, were calculated together. The amount of each individual HMO was calculated based on normalization to the internal standard (Supplementary Data 4). The relative abundance of each of the individual HMOs was determined by setting the sum of the 16 identified oligosaccharides as 100% total HMOs.

## Statistical analysis

No statistical method was used to predetermine the sample size. The investigators were not blinded to allocation during experiments and outcome assessment. Independent *t*-test was performed to test between groups when mentioned using the R function "t-test". Paired *t*-test was done between the percentages of infants in each cohort that had *BL. infantis* prior to 10 weeks and the percentage of infants that had detectable levels of *BL. infantis* after 10 weeks. Distances between consecutive infant gut microbiome samples and between breast milk HMO compositions were calculated using Bray-Curtis dissimilarity using the "vegdist" function from the vegan[65] (2.6–4) package. Correlation between the microbial population and HMO composition was performed using Pearson correlation with the "cor" R function. Adjusted *p*-values were calculated using "corr.test" from the psych package[70].

## Linear association models

The "Maaslin2"[71] R package was used to perform linear models in order to find associations between 16 HMOs and *Bifidobacterium* species and in the infant gut bacteria. The individual was set as a random factor to account for the effect of each mother-infant pair. In addition, "Maaslin2" was used to perform linear association models of age compared to *Bifidobacterium* species, adjusted to individuals.

## Reporting summary

Further information on research design is available in the Nature Portfolio Reporting Summary linked to this article.

## Data availability

The Human-filtered metagenomic sequencing data generated in this study has been deposited in the SRA database under BioProject PRJNA994433. Metadata of the cohort is provided in Supplementary Data 1. HMO quantification results are provided in Supplementary Data 4 and MetaPhlAn results in Supplementary Data 5.

## Code availability

Our tailored MetaPhlAn database is available on our GitHub page (https://github.com/yassourlab/MetaPhlAn-B.infantis/).

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

## Acknowledgements

This work was funded in part by the Azrieli Foundation grant for faculty fellows (M.Y.), by the Israel Science Foundation grant 2660/18 (D.E. & M.Y.) by the Waterloo foundation (D.E. & M.Y.), and by the Austrian Science Fund (FWF), under project number KLI 784 (E.J.K). M.Y. is the Rosalind, Paul and Robin Berlin Faculty Development Chair in Perinatal Research.

## Author contributions

D.E. established the cohort, generated the sequencing data, quantified the HMO abundance and performed all analyses. S.S. assisted with the experimental setup. E.J.K. guided and taught the HMO-quantification method. M.Y. guided the work. D.E. and M.Y. wrote the manuscript.

## Competing interests

The authors declare no competing interests.

## Additional information

Ethics declarations All mothers have agreed to participate in our study, which was approved by the Hebrew University's Institutional Review Board (IRB, approval number 20042021), and signed our consent forms for themselves and their infants. No compensation was provided to participants.

