## [Peer Review File · Nature Communications]

REVIEWER EXPERTISE

Reviewer #1. Bifidobacterium / HMOs / Infant microbiome / Mass Spec.

Reviewer #2. Metagenomics / Bioinformatics / Microbiome / Mother-infant.

REVIEWER COMMENTS

Reviewer #1 (Remarks to the Author):

This is a timely and interesting paper - there has long been a need a way to reliably separate *B. longum longum* and *B. longum infantis* from metagenomic data, so I was thrilled to see this method developed and validated against subspecies specific qPCR.

I am a bit confused by your subject and sample numbers. You have 21 mother-infant dyads, but 83 samples collected on the same day. Does this mean that you collected multiple samples from the same dyad on the same day? Or that you collected breastmilk and stool samples on the same day? And if you have paired samples, how did you wind up with an odd number of samples? This section needs more detail so if I am to understand what was done. A timeline indicating when dyads were enrolled in the study and when samples were collected would be very helpful in making sense of this section.

A timeline would also be helpful tracking levels of Bifidobacterium in individual infants, something that could be feasible with just 21 dyads. Some cultural context may also be helpful, for example, at what ages might infants start to attend daycare or otherwise have greater mixing in the community that might drive some of the changes discussed? If daycare typical begins at age 10 weeks, or some other shift in care patterns occurs at this point, it may explain the delayed colonization with *B. infantis*

I also am a bit confused by the statistical methods of this paper. Perhaps a detailed stats method section with an explanation of how the longitudinal nature of the data was dealt with would help?

Reviewer #2 (Remarks to the Author):

Ennis and colleagues present an interesting work focusing mainly on the *Bifidobacterium longum* subspecies and more specifically on the *B. longum* subsp. *infantis*. The Authors present their approach to survey these subspecies from metagenomic data by exploiting a new tailored marker database that they constructed to be used when profiling with MetaPhlAn, an existing tool for taxonomic profiling. Moreover, they report qPCR data validating their approach and an example application on a new cohort of infant gut microbiomes from Israel. Ennis and colleagues focus on the presence, relative abundance, and time shifts in *BL. infantis* and *BL. longum*, together with some comparison with *B. breve* and others. They moreover try to correlate maternal milk HMO composition with the *Bifidobacterium* dynamics in the infant gut, but they find none, as they state in their proposed title, thus suggesting that HMOs might not be the key and only factor driving the *Bifidobacterium* strain substitutions.

Despite the great importance of such work for the field, I have different major concerns:

1. The tailored database. It is not clear how the markers were built and tested to be subspecies-specific, except for qPCR validation which is great. More in detail:

a) Were the markers built by using only *BL infantis* and *BL longum* genomes or were other closely related species included? In Lines 364-368 of the Methods, there is a quick explanation but no data are reported. How did the Authors use Blastn to rule out other species? “not present at all in other subspecies”: which subspecies? How many markers were discarded for not being species- or subspecies-specific? More details and data are needed, including a test on other reference genomes of other subspecies and closely related species (e.g. *B. breve*) showing that these markers do not match them at all.

b) There are many more *BL infantis* markers (118) than *BL longum* markers (90) despite the larger number of genomes available for *BL longum*, which further support that there might be other species / subspecies mislabelled as *BL longum* in the reference genomes used.

c) Two out of 17 *BL infantis* reference genomes were discarded as probably mis-annotated. How did the Authors test for misannotation in *BL longum*? As shown by the Authors in Figure 3C, *BL longum* has a high variability, which may suggest that this indeed contains a mixture of other subspecies.

d) which genome completeness and contamination threshold did the Authors use for including a reference genome as reliable? if partial genomes are used, markers may be retained by mistake as they are not detected in a given subspecies just because the genomes are not complete.

e) as stated in Lines 120-121, "BL longum encompasses a broader range of strains referred to as "all other" strains": what does this sentence mean? Is for instance BL suillum/suis included as BL longum?

f) related to the previous point, do the qPCR primers used tell apart BL subspecies other than BL infantis and BL longum?

2. Some statistics are poorly supported or performed, including the following:

a) Lines 196-198: comparing BL infantis subspecies with B. breve (which is a species) of course will yield a lower diversity... Additionally, depending on how BL longum was defined, e.g. if it is "all other strains" as stated in Lines 120-121, of course it has a larger diversity. This comparison could also be removed as it adds very little to the story.

b) Lines 263-266: statistics suggesting that BL infantis prevalence varies across different cultures (which may well be) are done by including samples with different age distributions, with a US cohort with samples taken during the first 2 weeks of life and Swedish and Israeli cohorts with later timepoints. This test should be carried out by age-matched groups (e.g. all samples <2weeks), otherwise it is not meaningful at all to say it is linked to geography / culture. In addition, did the Authors control for the breastfeeding habits of the other two cohorts? Was the breastfeeding prevalence similar? Please provide data for homogeneous samples time- and feeding-wise.

c) Lines 271-273: High similarity between strains from same location: yes and no, as it seems that there are two sub-clusters with a mix of Swedish and Israeli samples, and clustering of the Israeli samples seems to be more due to individual than location. Also, were there siblings in the cohort?

d) overall, the only p-values reported are those for the validation through qPCR. There are no details on how the statistics were performed (if any), and many claims including the ones reported above do not mention statistical support.

Moreover, I also have some minor comments that need to be addressed before publication.

1. Lines 57-59: this sentence is a bit confusing. Why do Authors refer to "unknown sub-clade"? Maybe they mean a "not better-specified sub-clade"? or maybe "may represent multiple species"? Please clarify.

2. Lines 77-79: "the newly developed method" has never been mentioned before in the introduction, so please modify it with something like "the new method we propose here". In the same sentence,

“existing data in the literature” should probably be substituted with “data available in public repositories”?

3. Line 92: please specify how many infant stool and milk samples: is this 83 infant stool + 83 matched milk samples? It would help the reader to have clear numbers.

4. Did the Authors try to sequence metagenomes from milk as well? if so, how did this perform?

5. Line 100: “metagenomic classification” should probably be “taxonomic classification from metagenomes”?

6. Lines 190-192: how are “enriched transitions” defined? it seems that most of the changes affect the *BL longum*...

7. Line 204: “As expected” - please either remove or provide a reference on why this should be expected

8. Data availability: there are only 80 samples in the BioProject, were 3 discarded?

Figures:

Figure 1C is a duplicate of Supplementary Figure 2C?

Figure 6: Please mark the time point with the same scale (either weeks or months) for panels A, B, and C, otherwise it is difficult to compare.

REVIEWER COMMENTS

Reviewer #1 (Remarks to the Author):

This is a timely and interesting paper - there has long been a need a way to reliably separate *B. longum longum* and *B. longum infantis* from metagenomic data, so I was thrilled to see this method developed and validated against subspecies specific qPCR.

I am a bit confused by your subject and sample numbers. You have 21 mother-infant dyads, but 83 samples collected on the same day. Does this mean that you collected multiple samples from the same dyad on the same day? Or that you collected breastmilk and stool samples on the same day? And if you have paired samples, how did you wind up with an odd number of samples? This section needs more detail so if I am to understand what was done. A timeline indicating when dyads were enrolled in the study and when samples were collected would be very helpful in making sense of this section.

We thank the reviewers for their comments, and apologize for the confusion. We collected 83 pairs of samples over time from 21 dyads, each pair consisting of an infant stool sample and a maternal breast milk sample. Each pair was collected on the same day. Of these, 80 stool samples underwent metagenomics analysis and 50 breast milk samples underwent HMO quantification.

We clarified this in the text as well, in lines 93-98: "Our cohort consists of 21 mother-infant dyads with matched infant stool samples and breast milk samples collected on the same day. Altogether, we collected 80 stool samples and 50 breast milk samples together with the infant nutritional information and antibiotic treatments (Supplementary Figure 1, Supplementary Table 1)."

In addition, the timeline of all collected samples and the type of data generated for each sample appears in Supplementary Figure 1.

A timeline would also be helpful tracking levels of *Bifidobacterium* in individual infants, something that could be feasible with just 21 dyads.

We thank the reviewer for this comment. We have created a figure showing the mean relative abundance over time of the main *Bifidobacterium* species we focused on (attached below). However, we find this figure a bit confusing and perhaps even misleading. The mean relative abundance of *BL. infantis* starts at 0% and then rises to a maximum of 20%, however the presence of *BL. infantis* in most infants is bi-modal, either does not exist or has a high relative abundance.

We have attached the figure here and can add it as a supplementary figure if the reviewer thinks it is important and adds additional information.

Some cultural context may also be helpful, for example, at what ages might infants start to attend daycare or otherwise have greater mixing in the community that might drive some of the changes discussed? If daycare typical begins at age 10 weeks, or some other shift in care patterns occurs at this point, it may explain the delayed colonization with *B. infantis*

We added information regarding the timing of starting day care in Israel in the discussion and suggested that environmental factors may influence the shifts we see in the microbiome population as well.

In lines 343-345: “In addition, we can not rule out environmental effects that may contribute to these changes such as starting daycare which in Israel typically occurs at 15-26 weeks of age.”

I also am a bit confused by the statistical methods of this paper. Perhaps a detailed stats method section with an explanation of how the longitudinal nature of the data was dealt with would help?

We have added two sections in the methods (Statistical analysis and Linear association models, lines 496-514) explaining in detail all statistical tests used. In addition, we added in the text what statistical tests we used and the p-values.

Regarding associations between HMOs in mother milk and *Bifidobacterium* species in the infant gut, we have performed both Pearson correlation and linear regression models which were adjusted for individuals. In addition, we used linear models to look for associations between infant age and specific HMOs, using the longitudinal information of the data. We have explained this now in the Methods section.

Reviewer #2 (Remarks to the Author):

Ennis and colleagues present an interesting work focusing mainly on the *Bifidobacterium longum* subspecies and more specifically on the *B. longum* subsp. *infantis*. The Authors present their approach to survey these subspecies from metagenomic data by exploiting a new tailored marker database that they constructed to be used when profiling with MetaPhlAn, an existing tool for taxonomic profiling. Moreover, they report qPCR data validating their approach and an example application on a new cohort of infant gut microbiomes from Israel. Ennis and colleagues focus on the presence, relative abundance, and time shifts in *BL. infantis* and *BL. longum*, together with some comparison with *B. breve* and others. They moreover try to correlate maternal milk HMO composition with the *Bifidobacterium* dynamics in the infant gut, but they find none, as they state in their proposed title, thus suggesting that HMOs might not be the key and only factor driving the *Bifidobacterium* strain substitutions.

Despite the great importance of such work for the field, I have different major concerns:

1. The tailored database. It is not clear how the markers were built and tested to be subspecies-specific, except for qPCR validation which is great.

We thank the reviewer for this detailed comment. Following your comments we re-thought our selection of reference genomes used to create the markers and filtered the strains we used to contain only strains that are clearly *BL. longum* or *BL. infantis*. We then recreated the database and reran the tool on our data. Overall, our results have not changed.

We have added in the methods section a description regarding the selection process of marker genes. In addition, we explained our decision to exclude two *BL. infantis* reference genomes based on previous work (Diaz et al., 2021) and on supplementary figure 7 which shows the pangenome of all the strains we used and the two removed strains.

Full details appear in the *B. longum* subspecies quantification section in the Methods.

More in detail:

a) Were the markers built by using only *BL. infantis* and *BL. longum* genomes or were other closely related species included? In Lines 364-368 of the Methods, there is a quick explanation but no data are reported. How did the Authors use Blastn to rule out other species? “not present at all in other subspecies”: which subspecies? How many markers were discarded for not being species- or subspecies-specific? More details and data are needed, including a test on other reference genomes of other subspecies and closely related species (e.g. *B. breve*) showing that these markers do not match them at all.

See above, text was updated.

b) There are many more *BL. infantis* markers (118) than *BL. longum* markers (90) despite the larger number of genomes available for *BL. longum*, which further support that there might be other species / subspecies mislabelled as *BL. longum* in the reference genomes used.

c) Two out of 17 *BL infantis* reference genomes were discarded as probably mis-annotated. How did the Authors test for misannotation in *BL longum*? As shown by the Authors in Figure 3C, *BL longum* has a high variability, which may suggest that this indeed contains a mixture of other subspecies.

See above, text was updated.

d) which genome completeness and contamination threshold did the Authors use for including a reference genome as reliable? if partial genomes are used, markers may be retained by mistake as they are not detected in a given subspecies just because the genomes are not complete.

>90% completeness and <5% contamination. We now added this information to the Methods.

e) as stated in Lines 120-121, “*BL longum* encompasses a broader range of strains referred to as “all other” strains”: what does this sentence mean? Is for instance *BL suillum/suis* included as *BL longum*?

In the new database we included only reference genomes that were clearly annotated as *BL longum* or *BL infantis*. The sentence regarding the broader range of strains was removed.

We have added in the text explanation regarding *BL suis/suillum*, in lines 120-123:

“We chose to discard two subspecies of *B. longum*: *Bifidobacterium longum* subsp. *suis* and *Bifidobacterium longum* subsp. *suillum* since they are rarely found in humans, and a limited amount of reference genomes exist for these subspecies (Methods). “

And in lines 407-413:

“We decided not to include subspecies *BL suis* and *BL suillum* in our method due to the limited availability of reference genomes for these subspecies leading to difficulty to generate reliable marker genes. In addition, these subspecies were rarely reported in humans³⁴, thus are less relevant in our settings. Therefore it is important to note that using our method, *BL suis* and *BL suillum* can be misidentified as a different subspecies or alternatively identified as *B. longum* species with unclassified subspecies.”

f) related to the previous point, do the qPCR primers used tell apart *BL* subspecies other than *BL infantis* and *BL longum*?

The qPCR primers are subspecies specific, one set of primers for *BL infantis* and the other for *BL longum*. As mentioned before, we did not focus on other subspecies. We clarified this point in the Methods section.

2. Some statistics are poorly supported or performed, including the following:

a) Lines 196-198: comparing *BL infantis* subspecies with *B. breve* (which is a species) of course will yield a lower diversity... Additionally, depending on how *BL longum* was defined,

e.g. if it is "all other strains" as stated in Lines 120-121, of course it has a larger diversity. This comparison could also be removed as it adds very little to the story.

We thank the reviewer for pointing this out, our text was not clear enough. In these sentences we referred to the community diversity of the microbiome, namely the Shannon index of the abundances, which highlights how many other species are found in each sample, and their abundance. We do not discuss here the diversity of the strains, but rather the impact of being dominated by *BL. infantis* vs. *B. breve*, for example.

We changed the text to be clearer on this point, and highlighted the fact that when *BL. infantis* is found, it dominates the community at such high levels of relative abundance, leaving very little room for others to occupy, hence the low Shannon diversity. We have also added a section in the Methods about all the statistical analyses in the manuscript.

b) Lines 263-266: statistics suggesting that *BL. infantis* prevalence varies across different cultures (which may well be) are done by including samples with different age distributions, with a US cohort with samples taken during the first 2 weeks of life and Swedish and Israeli cohorts with later timepoints. This test should be carried out by age-matched groups (e.g. all samples <2weeks), otherwise it is not meaningful at all to say it is linked to geography / culture. In addition, did the Authors control for the breastfeeding habits of the other two cohorts? Was the breastfeeding prevalence similar? Please provide data for homogeneous samples time- and feeding-wise.

Due to the difficulty of comparing across cohorts with different time points and different cultural habitats we have removed this analysis from the paper.

Instead, we added more cohorts to our analysis and created a new figure (Figure 6E) comparing the prevalence of *BL. infantis* before and after 10 weeks of age within each cohort, highlighting the late colonization of *BL. infantis* in multiple geographical locations.

c) Lines 271-273: High similarity between strains from same location: yes and no, as it seems that there are two sub-clusters with a mix of Swedish and Israeli samples, and clustering of the Israeli samples seems to be more due to individual than location. Also, were there siblings in the cohort?

After adding more cohorts to our analysis we found that our original observation when looking just at 3 cohorts was not as accurate as we thought. We have created a new phylogenetic tree of *BL. infantis* looking at 9 different geographical locations (Figure 6F) and in lines 293-308 we discuss our findings.

No siblings were found in any of the cohorts tested.

d) overall, the only p-values reported are those for the validation through qPCR. There are no details on how the statistics were performed (if any), and many claims including the ones reported above do not mention statistical support.

We have added multiple statistical tests and mentioned them in the text together with p-values. For example, t-test was used to compare between the alpha diversity in the

different sample groups we looked at (Figure 3C). We added p-values for the Pearson correlation between HMOs in mothers milk and bacteria in the infant gut, and performed linear association models which were adjusted to individuals.

We have added two sections in the methods (Statistical analysis and Linear association models, lines 496-514) explaining in detail all statistical tests used.

Moreover, I also have some minor comments that need to be addressed before publication.

1. Lines 57-59: this sentence is a bit confusing. Why do Authors refer to “unknown sub-clade”? Maybe they mean a “not better-specified sub-clade”? or maybe “may represent multiple species”? Please clarify.

We meant “may represent multiple species”, this was fixed in the text.

2. Lines 77-79: “the newly developed method” has never been mentioned before in the introduction, so please modify it with something like “the new method we propose here”. In the same sentence, “existing data in the literature” should probably be substituted with “data available in public repositories”?

We have changed the wording, thank you.

3. Line 92: please specify how many infant stool and milk samples: is this 83 infant stool + 83 matched milk samples? It would help the reader to have clear numbers.

We apologize for the confusion. We collected 83 pairs of samples over time from 21 dyads, each pair consisting of an infant stool sample and a maternal breast milk sample. Each pair was collected on the same day. Of these, 80 stool samples underwent metagenomics analysis and 50 breast milk samples underwent HMO quantification.

We clarified this in the text, in lines 93-98: “Our cohort consists of 21 mother-infant dyads with matched infant stool samples and breast milk samples collected on the same day. Altogether, we collected 80 stool samples and 50 breast milk samples together with the infant nutritional information and antibiotic treatments”.

A timeline with all samples collected and whether metagenomic analysis and HMO quantification was performed can be found in Supplementary Figure 1.

4. Did the Authors try to sequence metagenomes from milk as well? if so, how did this perform?

We have tried multiple times to look at the milk microbiome using metagenomic sequencing. However, we believe it is difficult to achieve a metagenomic sequencing which is clean of contamination due to the very low biomass of bacteria found in breast milk.

We are currently using third generation sequencing (nanopore) to generate long reads of 16S-rRNA which might help us achieve the resolution we want but can not get with short read 16-rRNA sequencing.

5. Line 100: “metagenomic classification” should probably be “taxonomic classification from metagenomes”?

Thank you, we have fixed this in the text.

6. Lines 190-192: how are “enriched transitions” defined? it seems that most of the changes affect the *BL longum*...

We reconsidered this analysis and decided that it does not merit mentioning in the text. Thank you for pointing this out.

7. Line 204: “As expected” - please either remove or provide a reference on why this should be expected

We have added a citation.

8. Data availability: there are only 80 samples in the BioProject, were 3 discarded?

See answer to 3

Figures:

Figure 1C is a duplicate of Supplementary Figure 2C?

In Supplementary Figure 2C *B. longum* unclassified is added to the relative abundance of *BL. longum* found using the tailored MetaPhlAn. This improves the correlation coefficient, showing that usually *B. longum* unclassified is *BL. longum*. We can remove the figure if the reviewer thinks it is confusing.

Figure 6: Please mark the time point with the same scale (either weeks or months) for panels A, B, and C, otherwise it is difficult to compare

We have changed the figure so that all time points are in weeks.

Thank you for the detailed comments throughout the manuscript, we see clearly how the manuscript has improved.

REVIEWERS' COMMENTS

Reviewer #2 (Remarks to the Author):

I would like to thank the Authors for their effort in improving the manuscript's clarity and analysis. They went to great lengths to answer the detailed comments by both reviewers, and I really appreciate the quality of the updated work, including the new geographical analysis.

I have no further remarks.